# Using LANDSAT 8 and VENμS Data to Study the Effect of Geodiversity on Soil Moisture Dynamics in a Semiarid Shrubland

**Vladislav Dubinin** [1], **Tal Svoray** [2,3], **Ilan Stavi** [4] and **Hezi Yizhaq** [5,*]

[1] Earth and Planetary Sciences Department, Weizmann Institute, Rehovot 7610001, Israel; vladislav.dubinin@weizmann.ac.il
[2] Department of Geography and Environmental Development, Ben-Gurion University of the Negev, Beer-Sheva 8410500, Israel; tsvoray@bgu.ac.il
[3] Department of Psychology, Ben-Gurion University of the Negev, Beer-Sheva 8410500, Israel
[4] Dead Sea and Arava Science Center, Yotvata 88820, Israel; istavi@adssc.org
[5] Department of Solar Energy and Environmental Physics, Jacob Blaustein Institute for Desert Research, Ben-Gurion University of Negev, P.O.B. 653, Beer-Sheva 8410501, Israel
[*] Correspondence: hezi.yizhaq1@gmail.com

**Abstract:** Soil moisture content (SMC) is a limiting factor to ecosystem productivity in semiarid shrublands. Long-term droughts due to climatic changes may increase the water stress imposed on these lands. Recent observations demonstrate positive relations between geodiversity—expressed by the degree of soil stoniness—and SMC in the upper soil layers. This suggests that areas of high geodiversity can potentially provide a haven for plant survival under water scarcity conditions. The objective of this study was to assess the effect of geodiversity on the dynamics of SMC in semiarid environments, which so far has not been fully investigated. The optical trapezoid model (OPTRAM) applied to six-year time series data (November 2013–July 2018), obtained from LANDSAT 8 and highly correlated with field measurements ($R^2 = 0.96$), shows here that the SMC in hillslopes with high geodiversity is consistently greater than that in hillslopes with low geodiversity. During winter periods (December–March), the difference between the two hillslope types was ~7%, while during summer periods (June–September) it reduced to ~4%. By using the high-resolution spectral-spatiotemporal VENμS data, we further studied the geodiversity mechanism during summertime, and at a smaller spatial scale. The VENμS-based Crop Water Content Index (CWCI) was compared with the OPTRAM measurements ($R^2 = 0.71$). The Augmented Dickey–Fuller test showed that water loss in the high-geodiversity areas during summers was very small ($p$-value > 0.1). Furthermore, the biocrust index based on the VENμS data showed that biological crust activity in the high-geodiversity hillslopes during summers is high and almost stationary (ADF $p$-value > 0.1). We suggest that the mechanism responsible for the high SMC in the high-geodiversity areas may be related to lower evaporation rates in the dry season and high runoff rates in the wet season, both of which are the combined result of the greater presence of developed biocrusts and stoniness in the areas of higher geodiversity.

**Keywords:** drought; OPTRAM; biocrust; VENμS; LANDSAT 8

## 1. Introduction

Decreases in rainfall rates and increases in surface temperatures due to global warming processes may cause severe water deficits in semiarid ecosystems [1–3]. These ecosystems are composed of mosaics of self-organized patches of shrubs, bare soil, and herbaceous vegetation [4–7]. The main feedback in these semiarid landscape structures is between vegetation productivity and soil moisture

content (SMC). Direct rainfall and runoff subsidies are the main water sources and the limiting factors to sustain such ecosystems [8]. The bare soil areas, also known as the inter-patch spaces, play a critical role in regulating SMC in semiarid environments. These spaces are often covered by biological and physical crusts, which may limit water infiltration, therefore increasing the overland water flow that accumulates in the downslope vegetation patches [9–11]. Another important environmental factor, whose importance in regulating SMC was recently highlighted, is the stoniness level in the soil and vegetation patches. According to recent works, areas densely covered by rock fragments experience a shading effect, with a resultant lower evaporation rate [12,13]. The interactions between an ecosystem's physical and biotic components affect the soil–water budget and dynamics. Consequently, these interactions play a major role in the durability and resilience of semiarid shrubland ecosystems to prolonged droughts and climatic changes [12,14,15].

Geodiversity is defined as the diversity of geological (rock fragments and stones), geomorphological (physical processes and landforms), and soil attributes [16]. Recent studies in the Israeli Negev region reported on high-geodiversity hillslopes, with a thin soil layer, approximately 30% stoniness in the soil profile, and ~25% rock fragment cover of the ground surface [14]. Low-geodiversity hillslopes exist in close proximity to these hillslopes, and are defined with a thick soil layer and no stoniness in the soil profile nor on the ground surface [17]. The soil thickness across the study site seems to be determined by site-specific drainage basin characteristics and long-term erosional processes [18]. It was reported that the high-geodiversity hillslopes help shrubs in coping with water deficits during prolonged droughts [15,17,19,20]. The positive impact of stoniness on the soil–water content could be simultaneously affected by several mechanisms, including (1) the formation of a water film, which wraps the rock fragments in the soil profile [21]; (2) water storage in the pores of rock fragments across the soil profile [21]; and (3) rock fragments on the ground, which affect the formation of water overland flow [22]. Regardless, the soil moisture regime is also affected by the thickness of the soil down to the underlying bedrock, as well as by the bedrock's weathering and cracking [17]. Specifically, previous studies showed that SMC in high-geodiversity areas is greater than in low-geodiversity areas. Nonetheless, these findings are based on a limited amount of field measurements during the rainy season from October to May, or on numerical simulations of vegetation in dry ecosystems by using a mathematical model. Moreover, the SMC measurements in these studies are limited to relatively small spatial scales [15,17,23].

Studying temporal variations in SMC over large spatial scales and at a high spatiotemporal resolution is, therefore, needed for better understanding the role of geodiversity in determining vegetation dynamics in semiarid ecosystems. Measurements should allow the elucidation of the role of geodiversity or spatial variability in regulating the water-cycle-related processes taking place between the atmosphere and the ground, such as evaporation, infiltration, and overland flow generation [24,25]. However, if the data obtained on-site are limited, the link between geodiversity and the water cycle is not fully understood. Therefore, the use of remote sensing data at a high spatiotemporal resolution can considerably improve our understanding of the role of geodiversity in increasing SMC in specific patches in semiarid environments.

Remote sensing data can be used to assess SMC in the soil surface layer by various means [26–28]. SMC at the ground surface is a fundamental state parameter, which refers to the water exchange at the land–atmosphere interface through water infiltration into the soil, runoff and the spatial redistribution of surface water, evaporation, and transpiration [25,29]. Two remote sensing methods are frequently used for SMC assessment: (1) using known water absorption bands from soil and vegetation based on the reflectance in the short-wavelength spectral band of 0.35–2.5 μm [30–32]; and (2) using the combined data of short-wavelength radiation and thermal infrared radiation in the band of 1.4–3.5 μm, which is known as the "trapezoid method" [33,34]. However, current satellite data provide a limited spatial resolution of thermal infrared radiation products; thus, methods that are based on shortwave radiation are preferred for SMC measurement [35].

The optical trapezoid model (OPTRAM), which was introduced by Sadeghi et al. [36], combines short-wavelength radiation with the Kubelka and Munk (KM) model in the "trapezoid method" to estimate SMC. The KM model is a physically based model that utilizes the relationships between the short-wavelength infrared (SWIR) and dielectric properties of water to estimate SMC [37]. "Trapezoid methods" are based on the inverse relationships between vegetation cover, which can be estimated by vegetation indices (VIs), e.g., the normalized difference vegetation index (NDVI), and land surface temperature [31]. These relationships are estimated through two linear equations for a low and high land surface temperature, thus creating a trapezoidal area, while each point inside the area is correlated to the SMC (see Figure 1). The KM model allows the land surface temperature to be replaced by a parameter based on SWIR reflectance.

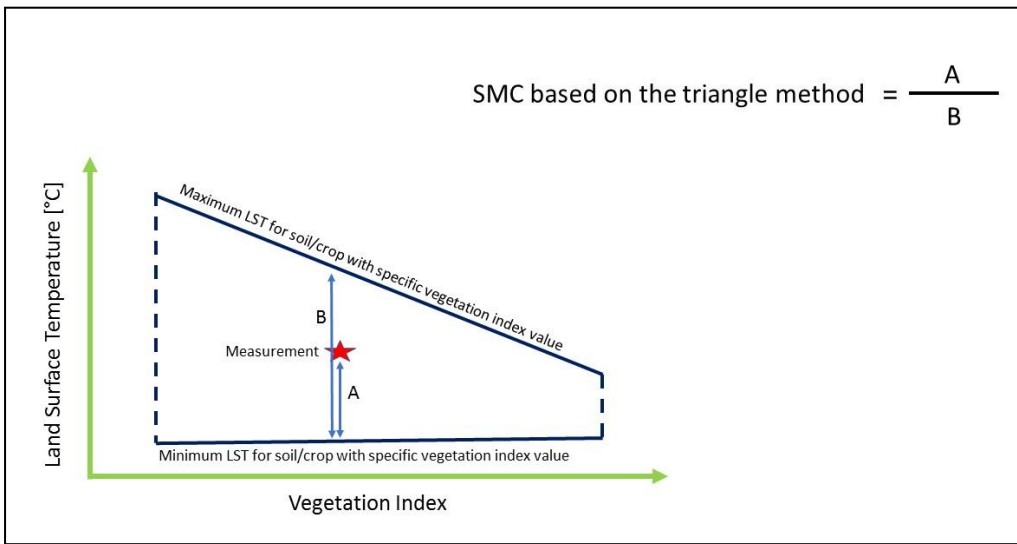

**Figure 1.** Schematic triangle graph of the relation between vegetation indices and land surface temperature (LST). The area between the lower and the upper borders represents various soil moisture content (SMC) values. (1) The red asterisk represents the measurement of SMC based on the triangle method (see the equation in the figure); (2) arrow A represents the distance in LST from the minimum LST of the soil/crop with a specific vegetation index value; and (3) arrow B represents the distance between the minimum and maximum LST values for a specific vegetation index value. LST can be replaced by transformed reflectance in the SWIR spectral region.

LANDSAT 8, equipped with the Operational Land Imager (OLI) instrument, provides VIs and SWIR reflectance products with a spatial resolution of 30 m, and with a revisit time of 16 days. This may suffice to estimate hillslope-scale SMC, but not to assess SMC in small patches of a few meters [38,39]. The new Vegetation and Environment monitoring on a New Micro-Satellite (VENμS) is a near polar sun-synchronous orbit microsatellite that allows the assessment of vegetation and soil at a higher spatial resolution of 5.3 m, a revisit time of two days, and at 12 spectral bands (0.42–0.91 μm). These capabilities can provide additional data on the effects of small-scale geodiversity on canopy water content, as well as on factors that affect the dynamics of canopy water content, such as biocrust cover [40].

The effects of biocrusts on the SMC balance in semiarid environments have been thoroughly and widely studied [9,41,42]. The results show that, on the one hand, cyanobacteria-dominated biocrusts, in particular, prevent infiltration [41] and contribute to runoff [42]. On the other hand, biocrust on sandy soils allows high evaporation from the soil due to low albedo [43,44]. However, biocrusts and physical crusts decrease the evaporation from the deep soil layers in non-sandy soils [45]. Findings by Kidron and Aloni [44] showed that biocrusts assist the survival of shallow-rooted perennials, but increase the mortality rates of other perennials, as well as of annuals. Using the VENμS satellite data is expected to allow a better understanding of the role of biocrusts in regulating SMC.

Our aim here is to assess the effect of geodiversity, expressed by thickness of the soil and degrees of stoniness, on regulating the seasonal dynamics of SMC in a semiarid site in Israel's northwestern Negev Desert. The remotely sensed data used here include time series of LANDSAT 8 and VENμS. The specific operative objectives were (1) to test SMC predictions using OPTRAM applied to LANDSAT 8 images against field measurements; (2) to study SMC spatiotemporal dynamics predicted by OPTRAM; and (3) to study the effect of geodiversity on canopy water content and biocrust activity during the dry season using the high spatiotemporal resolution of the VENμS images. We hypothesized that in plots of high geodiversity, SMC would be higher than in plots of low geodiversity, and that the time series analysis of LANDSAT 8 and VENμS images would be able to show this at different observational scales.

## 2. Materials and Methods

### 2.1. Study Site

This study was carried out in the Sayeret Shaked Park Long Term Ecological Research (LTER) Station (Figure 2), located in the northwestern Negev Desert, Israel (31°27′ N, 34°65′ E). The wet season in this area occurs between November and March, with rainfall events non-uniformly distributed both annually and inter-annually. The mean annual rainfall between 1992 and 2016 was 165 mm. Mean daily temperature ranges between 12 °C in February and 26 °C in July [15]. The LTER site extends over an area of 20 ha, and is fenced to prevent livestock access. The site is characterized by a hilly terrain, with hillslope inclination ranging between 3° and 6°. Lithology is dominantly composed of chalk, and the soil is loessial calcic xerosol, with a sandy loam to loamy sand texture [17].

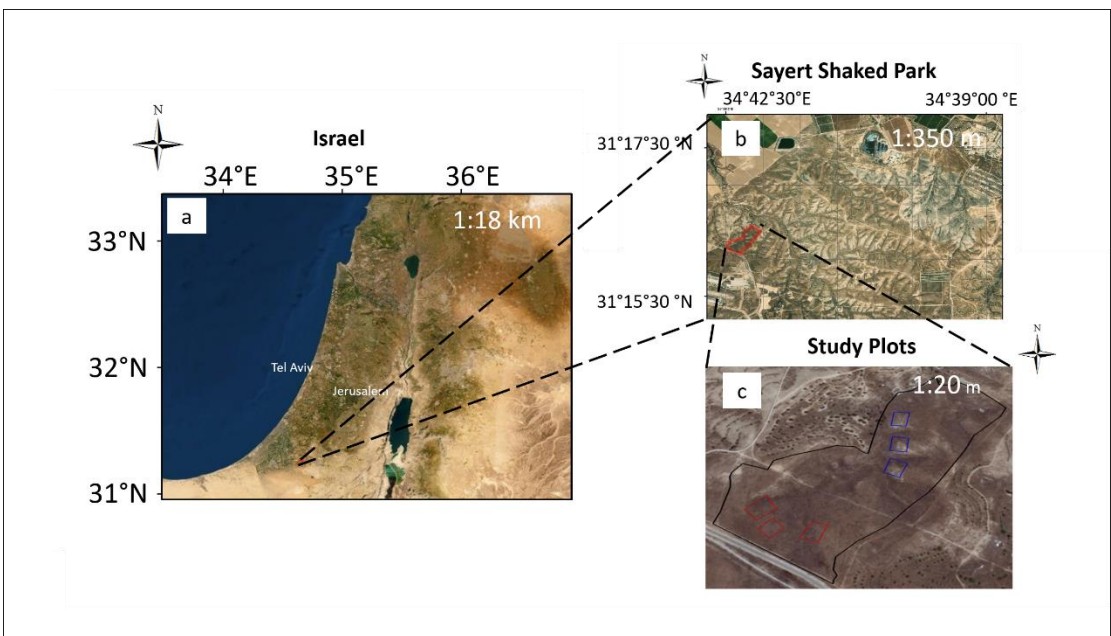

**Figure 2.** Maps of the study area of the Sayeret Shaked LTER. Map (**a**) presents the Sayeret Shaked LTER location in Israel, Map (**b**) presents the location of the study area, and Map (**c**) presents the location of the studied plots in the Sayeret Shaked LTER, where the red plots represent low-geodiversity hillslopes and the blue plots represent high-geodiversity hillslopes (Google Earth 2020).

The study was carried out in six hillslopes, on each of which, a plot of 20 × 20 m was delineated (Figure 2c). In terms of the geodiversity components (soil thickness, stoniness in the soil profile, and rock fragments cover) [15] of these six hillslopes, three are characterized by high geodiversity, while the other three are characterized by low geodiversity. The high-geodiversity hillslopes have a thin soil layer of about 0.1 m, high stoniness of about 35%, and are sparsely covered with herbaceous vegetation (29.7%) and live shrubby vegetation (24.9%) [46]. The hillslopes with low geodiversity

are characterized by a thick soil layer (depth > 1 m), free from rock fragments, and a dense cover of herbaceous vegetation (92.5%) and mostly dead shrubs (live 2.2%, dead 5.3%).

In order to minimize the impact of aspect orientation, the six chosen hillslopes where those with as much a similar azimuth as possible; also, in order to negate the impact of catenary positioning, all of the study plots were delineated in the hillslopes' backslope unit [17]. A recent study reported on considerable differences in soil properties between the two types of hillslopes (as shown in Figure 3). Of particular relevance is the distinct soil texture, which was found to be considerably finer in the high-geodiversity hillslopes than that in the low-geodiversity ones [46]. Details about the specific species of the shrubby vegetation in the study area can be found in [19].

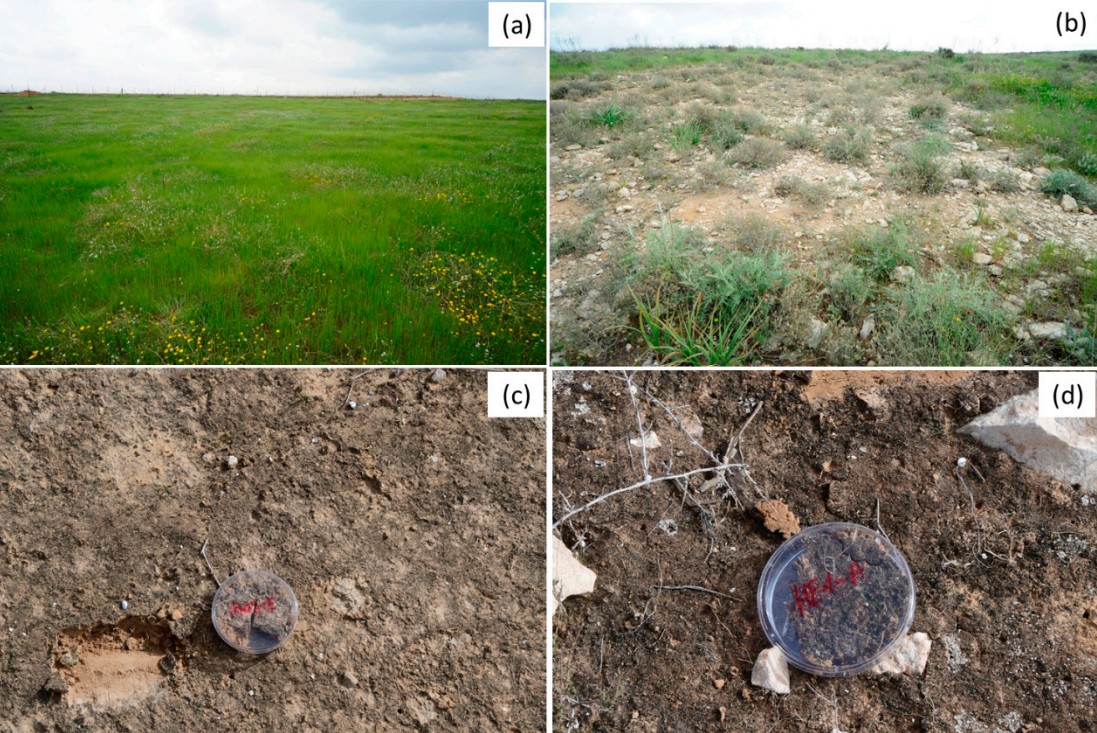

**Figure 3.** (**a**) A typical non-stony, low-diversity hillslope in the study region (spring 2017). Note the very dense cover of herbaceous vegetation. (**b**) A typical stony, high-geodiversity hillslope in the field study covered by plenty of shrubs and abundant rock fragments. (**c**) The cyanobacteria biocrust in the low-geodiversity hillslope. (**d**) The more developed biocrust in the high-geodiversity hillslope indicated by a darker color due to presence of moss and lichens.

*2.2. Data Processing*

2.2.1. LANDSAT 8 Imagery

LANDSAT 8 (pixel size of 30 m and revisit time of 16 days) has 12 spectral bands. Of them, two are sensitive to the SWIR spectral region (band 6: 1.57–1.65 μm and band 7: 2.11–2.29 μm); one is sensitive to the near infrared spectral region (band 5: 0.85–0.88 μm); and another is sensitive to the red spectral region (band 4: 0.64–0.67 μm). These features make the LANDSAT 8 suitable for applying the OPTRAM model.

The LANDSAT 8 images used in this work were acquired between November 2013 and July 2018, with cloud cover of <40% (see more details on the LANDSAT 8 images used here in Table S1 in the Supplementary Information). Image downloading was applied by using the website platform of the U.S. Geological Survey (USGS: https://earthexplorer.usgs.gov/). The selected images were geometrically corrected by using Ground Control Points (GCP) and digital elevation model (DEM) data. Radiometric

correction for each image was applied in order to convert the digital numbers (DNs) to the top of the atmosphere reflectance (TOA) by using the attached Metadata and Equation (1) (USGS):

$$\text{TOA} = \frac{M_\rho * DN + A_\rho}{\cos(90 - \theta_{SE})} \tag{1}$$

where $M_\rho$ is a band-specific multiplicative rescaling factor; DN is a digital number of pixel I; $A_\rho$ is a band-specific additive rescaling factor; and $\theta_{SE}$ is a local sun elevation angle. Atmospheric correction was performed using the Dark Object Subtraction 4 method [47] with the function calcAtmosCorr in the satellite package [48] in R software [49]. The corrected images were cropped with the study area boundaries.

### 2.2.2. OPTRAM Model for Estimating SMC

According to Sadeghi et al. [36], OPTRAM includes four computation steps: (a) transformed reflectance computation; (b) calculation of NDVI; (c) estimation of the minimal and maximal borders by a quantile regression of transformed reflectance and NDVI; and (d) calculation of OPTRAM using the normalization of transformed reflectance by the minimal and maximal borders. Here, we describe the OPTRAM computations:

a.　Transformed reflectance computation

The transformed reflectance (r), according to Sadeghi et al. [29], was defined according to Equation (2):

$$r = \frac{(1-R)^2}{2R} \tag{2}$$

where R is the SWIR reflectance of the atmospherically and radiometrically corrected images. Here, we only used the reflectance data of band 6 (1.57–1.65 μm) for stage one. This is because in our case band 6 was less noisy than band 7 (2.11–2.29 μm) [36].

b.　Calculation of NDVI

For the vegetation cover component of OPTRAM, we used, as a proxy, the frequently used NDVI, as presented in Equation (3):

$$\text{NDVI} = \frac{\text{band 5} - \text{band 4}}{\text{band 5} + \text{band 4}} \tag{3}$$

where band 4 and band 5 represent the reflectance in the red and the NIR spectral regions of LANDSAT 8, respectively.

c.　Estimation of the minimal and maximal borders

Both the transformed reflectance (r) and NDVI were used to determine the minimal and maximal borders of the trapezoid area (Figure 1), using the linear equations presented in Equations (4) and (5):

$$r_{min, i} = a + bNDVI_i \tag{4}$$

$$r_{max, i} = c + dNDVI_i \tag{5}$$

where NDVIi is the value of NDVI in pixel i; and rmin,i and rmax,i represent 5% and 95% of the transformed reflectance (r), respectively. The coefficients a, b, c, and d are the linear regression coefficients of the lowest 5% of the transformed reflectance (r) data for the minimal border, and the highest 5% of the transformed reflectance (r) data for the maximal border. Here, we used the "rq" function of the "quantreg" package [50] in R for creating quantile linear regressions of the abovementioned equations (Equations (4) and (5)).

d.　　Calculation of OPTRAM

Finally, the transformed reflectance (r) of each pixel (i) was normalized to values varying between 0 and 1 using Equation (6):

$$OPTRAM_i = \frac{r_{max,i} - r_i}{r_{max,i} - r_{min,i}}$$

(6)

Equation (6) represents the OPTRAM values that are expected to be highly correlated with the SMC measurements [36].

2.2.3. Field Measurements

For three years (2015–2018), between December and May, 19 field campaigns of SMC measurements were conducted by TDR probes with a 7.6-cm length [17]. The measurements were taken using a time-domain reflectometer (TDR Spectrum Technologies©). In each plot, measurements were taken from 10 different spots, of which five were under shrubs and five in the bare soil between shrubs (here, we use the averages of these 10 measurements). Rainfall data were taken from November 2013 to July 2018 from a nearby meteorological station in the Sayeret Shaked Park LTER. For calibrating of the TDR readings, representative soil cores (of 100 mL volume each) were obtained, and their soil was put in a drying oven for determining volumetric moisture content [17,51,52].

2.2.4. VENµS Imagery and Vegetation Indices (VIs)

Level 2 of the VENµS images provides radiometric, geometric, and atmospheric corrections, and it is available from June 2019 (see list of VENµS images used in this research in the Supplementary Information, Table S2). The radiometric correction applied in Level 2 is based on simultaneous nadir observations (SNO) and Sentinel-2 satellite data [53]. The geometric correction is based on the same observation angle of each site and ground validation, resulting in a correction of less than 3 m per pixel (http://www.bgu.ac.il/BIDR/research/phys/remote/03-Venus.htm). The atmospheric correction is based on a Maccs-atcor joint algorithm (MAJA) atmospheric correction [54].

VENµS is a micro-satellite that was designed to monitor plant dynamics and status for agricultural and environmental purposes and, therefore, its spectral bands were designed to detect the signature of vegetation with high density along the red edge. VENµS has 12 spectral bands that are sensitive to the spectral region of 0.42 to 0.91 µm, where four of these bands are located on the red-edge region, which is sensitive to vegetation condition. Nevertheless, VENµS does not provide spectral bands in the SWIR region and, therefore, it cannot be used to measure SMC as was done using LANDSAT 8. However, band 12 (0.89–0.93 µm) is correlated to canopy water content but also to the SMC, as will be shown later.

The Crop Water Content Index (CWCI) is based on the water band index (WBI: [55]), using reflectance measurements in the spectral regions of 0.970 µm and 0.900 µm. The CWCI (Equation (7)) was used here for estimating the desert plant water concentration and was tested against OPTRAM by using a linear regression. Because vegetation canopy water content depends on SMC, particularly in semiarid environments, it is expected that a high correlation will occur between the two indices.

$$CWCI = \frac{\rho_{0.865} - \rho_{0.910}}{\rho_{0.865} + \rho_{0.910}}$$

(7)

where $\rho_{0.865}$ and $\rho_{0.910}$ are the reflectance values in the 0.865 µm and 0.910 µm spectral regions.

VENµS data allow the prediction of additional factors that can explain the relatively high SMC in the high-geodiversity hillslopes. One of these factors is the biological crust cover, which can decrease evaporation and implies high SMC in the summer because the survival of biogenic crust, as well as

other vegetation formations, depends on SMC in semiarid regions [11,45]. Karnieli in [40] introduced the crust index, which is correlated with biocrust cover in arid climate zones and is presented in Equation (8):

$$\text{Crust Index} = \frac{\rho_{0.620} - \rho_{0.443}}{\rho_{0.620} + \rho_{0.443}} \tag{8}$$

where $\rho_{0.620}$ and $\rho_{0.443}$ represent the reflectance in 0.620 and 0.443 μm. The crust index was tested by Karnieli [40] and Rodríguez-Caballero et al. [56] to investigate biological crust functioning in areas with environmental conditions similar to the current research's study area. The crust index successfully detected biocrust activity; however, the index's time dynamics was not yet tested.

### 2.2.5. Statistical Analysis

The SMC estimations, based on LANDSAT 8 data, were used to create the mean values for the time series from November 2013 to July 2018 for both high and low geodiversity plot types. The time series were compared using boxplots and *t*-tests. The time series trends were computed using the moving average of the three nearest points. The measurements were calibrated using the field data of the period from 2013 to 2018, while the calibration operation was applied using Ordinary Least Squares (OLS).

Time series based on VENμS satellite data, i.e., CWCI and the crust index, were analyzed with a *t*-test and an Augmented Dickey–Fuller test (ADF: [57,58]), which tests the existence of temporal trends. In the case of no statistically significant outputs, the temporal trend is explained by a horizontal line with a zero slope; otherwise, there is a more complicated behavior of the tested VIs. The ADF test was applied to the data using the "adf.test" function in the "tseries" package in R software [59].

### 2.2.6. Summary of the Main Method of the Work

In this research, we aimed to assess and understand the effect of geodiversity on soil moisture content (SMC). For this, we use two remote sensing platforms: LANDSAT 8 and VENμS satellites. We estimate SMC using the optical trapezoid model (OPTRAM) using LANDSAT 8, and the Crop Water Content Index (CWCI) using VENμS high spatial resolution images. To understand the role of biocrusts in SMC dynamics, biocrust activity was estimated by calculating a crust index from VENμS images. The different methods used in this work to study the effect of geodiversity on the SMC dynamics are schematically presented in Figure 4.

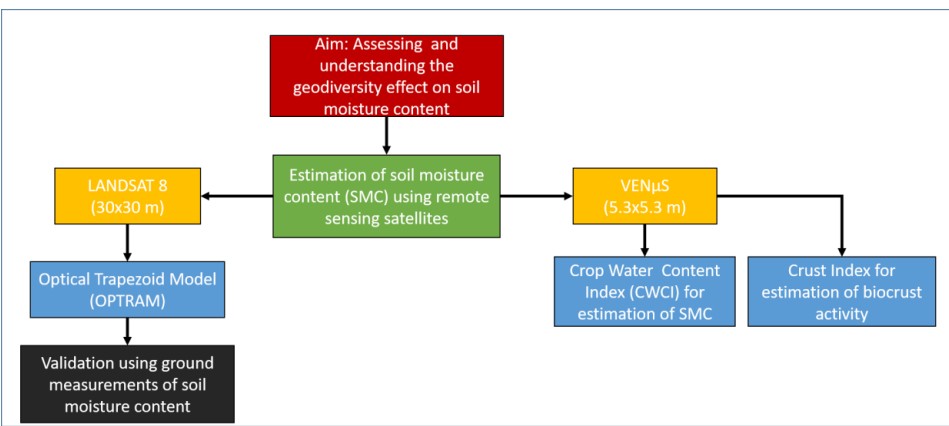

**Figure 4.** Flow chart of the main research methods. In this research, we aimed to assess and understand the effect of geodiversity on soil moisture content (SMC). For this, we used two remote sensing platforms: LANDSAT 8 and VENμS satellites. We estimated the SMC using the optical trapezoid model (OPTRAM) using LANDSAT 8, and the Crop Water Content Index (CWCI) using VENμS high spatial resolution images. To understand the role of biocrusts in SMC dynamics, biocrust activity was estimated by calculating a crust index from VENμS images.

## 3. Results

### 3.1. OPTRAM Images

The OPTRAM-based SMC predictions are shown in Figure 5. During the five-year period, the SMC for the high-geodiversity hillslopes was higher than that in the low-geodiversity hillslopes.

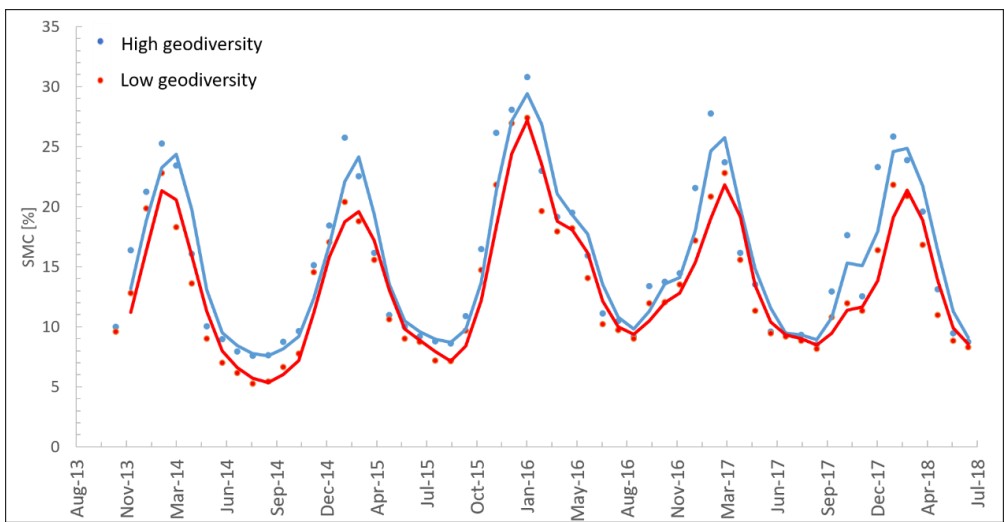

**Figure 5.** The SMC time series calculated by OPTRAM from 2013 to 2018 (each series is an average of three plots). The points represent the OPTRAM measurements, and the curves represent the time series of the hillslope types using the moving average.

The boxplot in Figure 6 and *t*-test verify this trend (i.e., *p*-value < 0.05), confirming that the SMC in the high-geodiversity hillslopes is higher. SMC depends on intra-annual changes, e.g., air temperature and, mainly, rainfall amount [60]. As a result, SMC was greater in the winter season (December–March) than in the summer.

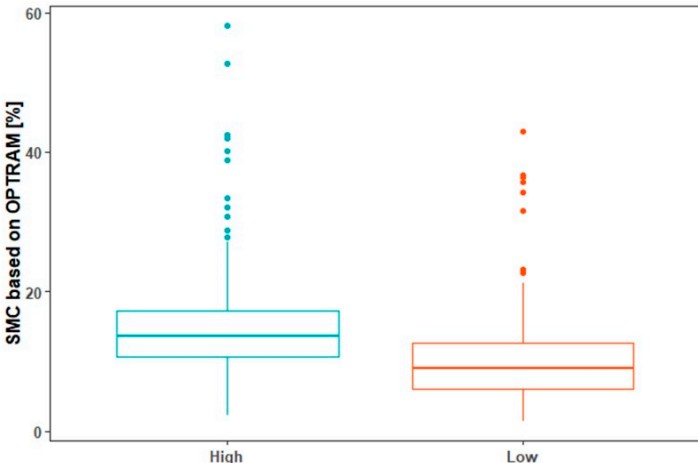

**Figure 6.** Boxplot comparison between the high-geodiversity hillslopes (pale blue boxplot) and the low-geodiversity hillslopes (red boxplot). The SMC was calculated by OPTRAM from 2013 to 2018. The difference in the mean SMC values between the two hillslope types is statistically significant (*p*-value < 0.05).

Figure 7 is a combined graph of the inter-annual distribution of rainfall in the Sayeret Shaked LTER during the six-year period of 2013–2018 (cumulative rainfall per month) and the SMC values based on OPTRAM for the two hillslope types with a mean measurement per month. The trends in

SMC and rainfall are similar; however, there is a small shift of one month between the peaks of SMC and rainfall amount. This shift is a result of LANDSAT 8's limited temporal resolution, which provides only two images per month. In the case of a cloudy image, OPTRAM measurements are based only on the partial data of non-cloudy images. As a result, we have gaps in the OPTRAM time series that can be as long as one month. No significant changes were observed between the years; however, the SMC in 2015 was a little greater than that in the other years, which is reasonable due to the high amount of rainfall in that year (225 mm; [15]). The difference in SMC values between the hillslopes is larger in the winter (~7%) and much smaller during the dry summer months (~4%).

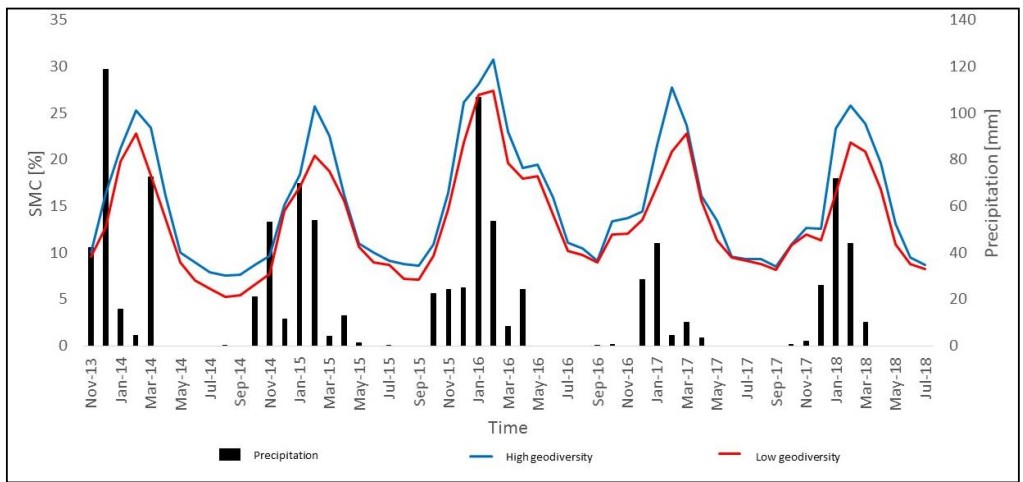

**Figure 7.** Precipitation and SMC monthly values from November 2013 to July 2018. The blue and the red curves present the SMC time series using the moving average, while the black barplots present the monthly cumulative amount of precipitation.

Figure 8 shows the results of the SMC field measurements (TDR) in the study plots using an electrode with a length of 7.6 cm. The overall trend is similar to the OPTRAM results. Both data series show a positive effect of geodiversity on the SMC, i.e., greater values in the high-geodiversity hillslopes. The maximum peak of soil moisture is in February, in agreement with the SMC measurements based on LANDSAT 8 data. Note that the SMC is always larger in the high-geodiversity hillslopes. The correlation between OPTRAM and the field measurements was very high ($R^2$ = 0.9556; see Figure 9).

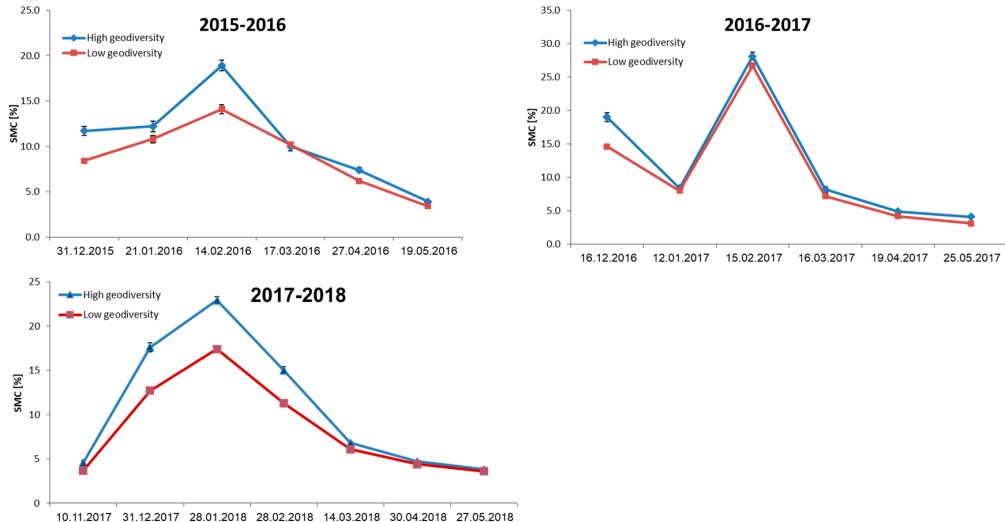

**Figure 8.** Field measurements of SMC in the high- and low-geodiversity hillslopes during three rainy seasons; heterogeneous hillslopes in blue and homogenous hillslopes in red.

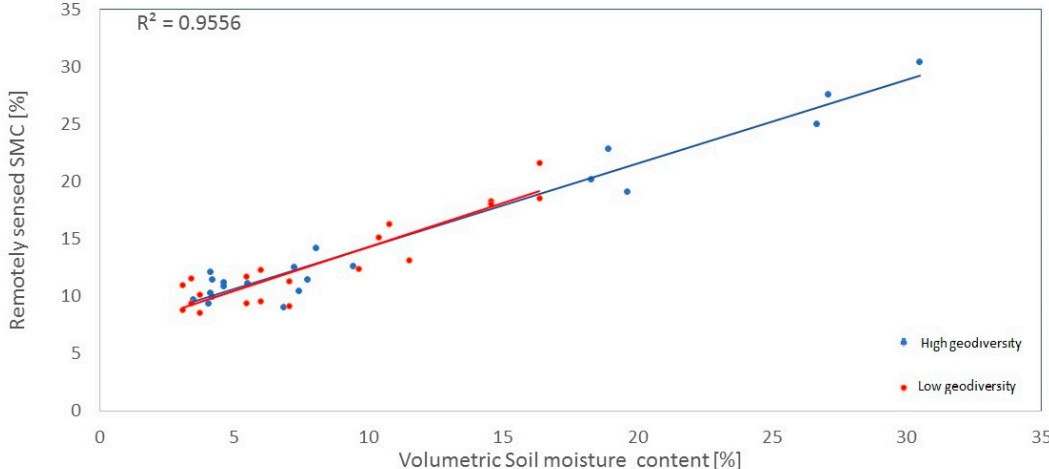

**Figure 9.** Validation of remotely sensed measurements of SMC with SMC field measurements. The $R^2$ value fits to all measurements in the heterogeneous hillslopes (blue) as well homogenous hillslopes (red).

### 3.2. VENµS Images

Figure 10 shows that vegetation canopy water content, expressed by the CWCI index, in the patches in the high-geodiversity hillslopes is greater than that in the low-geodiversity hillslopes (*t*-test, *p*-value < 0.05). The SMC trend in the high-geodiversity hillslopes was relatively stationary, while the trend in the low-geodiversity hillslopes shows a decrease in water content, with a minimum in August. The low-geodiversity hillslopes are covered by annuals as most of the perennials died due to prolonged droughts (the last one in 2008–2009, see Figure 1 in Yizhaq et al. [15]). In August, which is the at end of the summer, the annuals are almost dry so the CWCI is at its lowest value. On the other hand, the high-geodiversity hillslopes covered with perennial shrubs are less affected by the heat of the summer and the CWCI does not change significantly. The ADF test yielded significant results for the high-geodiversity hillslopes, i.e., temporal stationarity (*p*-value = 0.01), while the test was non-significant for the low-geodiversity hillslopes. In addition, the correlation between CWCI and OPTRAM is relatively high, as shown in Figure 11 ($R^2$ = 0.71). Thus, the correlation between the SMC and the canopy water content is clearly shown in our remotely sensed data.

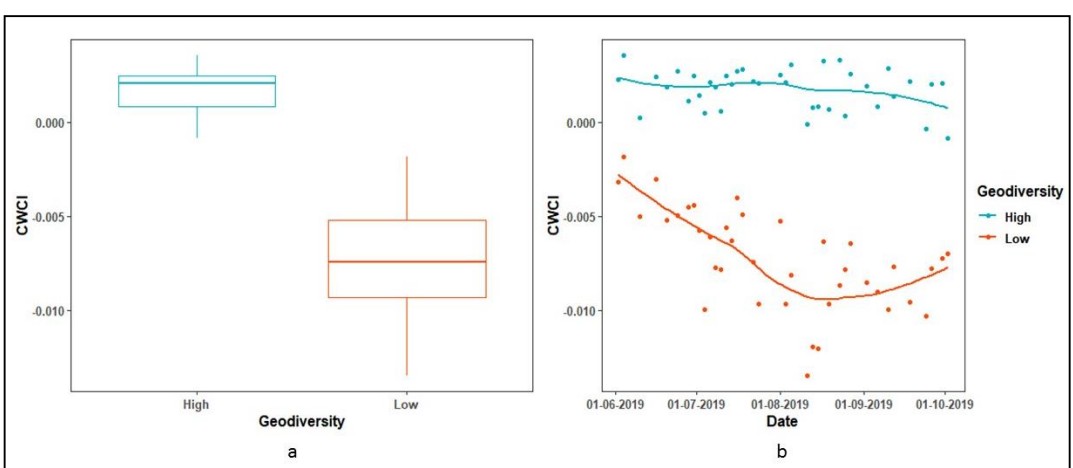

**Figure 10.** Crop Water Content Index (CWCI) time series analysis. Graph (**a**) represents the CWCI values in both types of hillslopes. Graph (**b**) shows the CWCI time series with the trend, using the LOESS method.

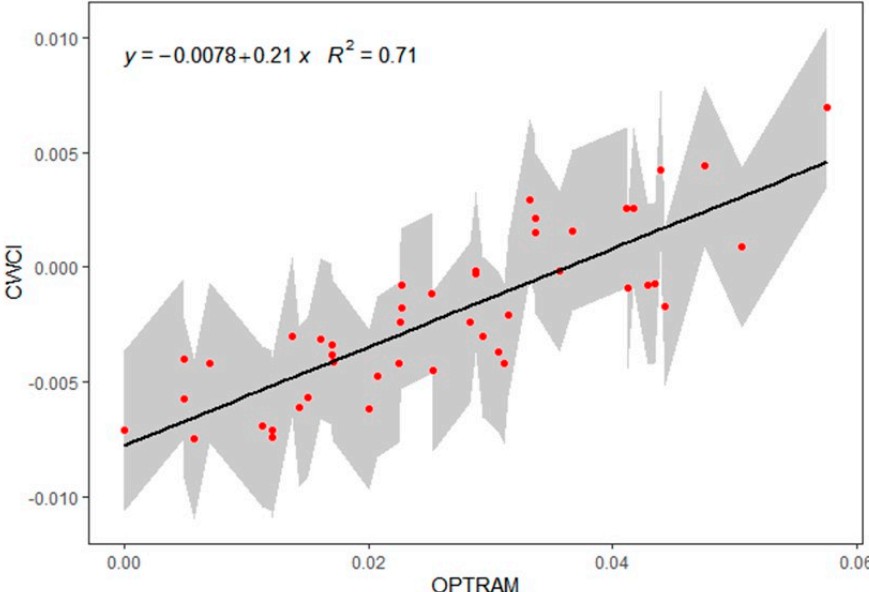

**Figure 11.** CWCI as a function of OPTRAM. The red points present the data from the same day of LANDSAT 8 (used for calculation of OPTRAM) and VENμS (used for calculation of CWCI). The line is a linear regression that is described by the equation CWCI = −0.0078 + 0.21*OPTRAM, and $R^2$ = 0.71. The shaded area presents the standard error in the linear regression.

The crust index results (Figure 12) show that the biocrust cover is more developed in the high-geodiversity hillslopes, and this was validated by a *t*-test (*p*-value = 0.0224). Nonetheless, the trend that appeared in both hillslope types was not stationary (*p*-value > 0.05). The ADF test results for the high-geodiversity hillslopes were almost statistically significant (*p*-value = 0.0753). The average range (i.e., without extreme points) of the biogenic crust index was between 0.33 and 0.36. The fluctuations in this index's results could be due to variability in the spectral reflectance caused by changes in atmospheric conditions. The spatial distribution within the plots of both indices between the beginning of summer (2 June 2019) and beginning of autumn (30 September 2019) is shown in Figure 13. This figure shows that the values of the CWCI and crust index in the high-geodiversity hillslopes (the three upper plots) are higher than those in the low-geodiversity hillslopes (the three bottom plots).

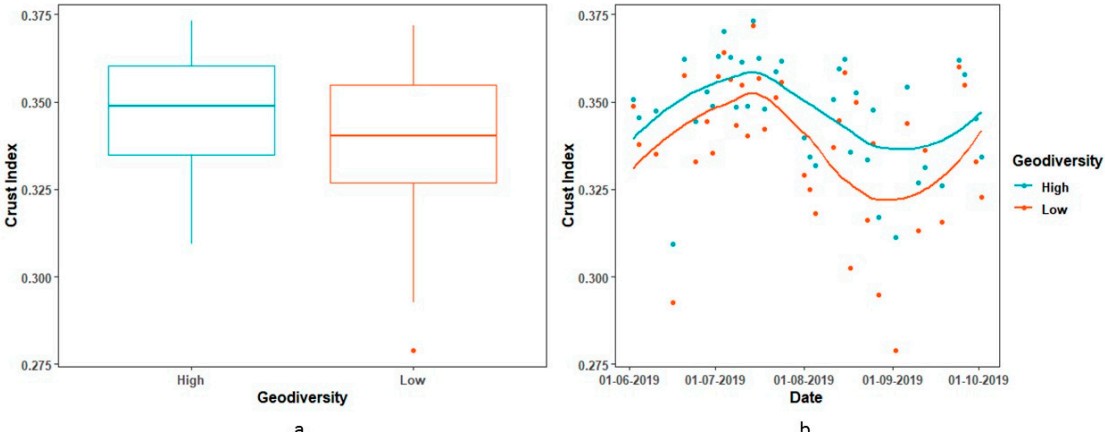

**Figure 12.** Crust index time series analysis. Graph (**a**) represents the crust index values in both types of hillslopes. The graph of the crust index is larger in the high-geodiversity hillslope than the low-geodiversity hillslope. Panel (**b**) shows the time series of the crust index with the fitted trend using the LOESS method from June to October 2019.

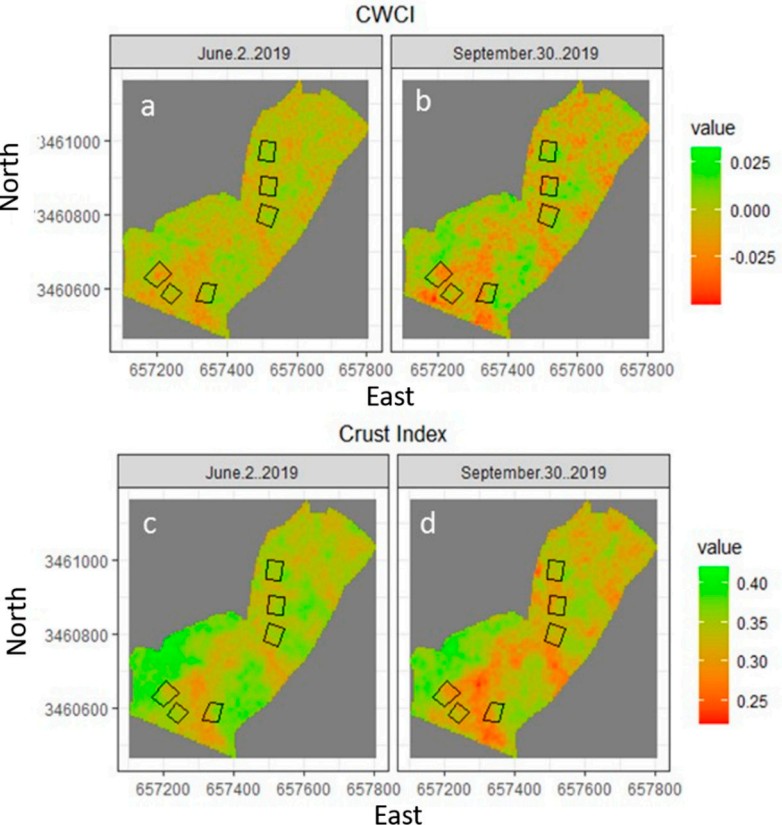

**Figure 13.** The CWCI (**a**,**b**) and crust index (**c**,**d**) spatial distribution from 2 June 2019 and 30 September 2019. It can be seen than (1) the top panels present that the CWCI values are higher in the high-geodiversity hillslopes (the three top plots) in the periods at the beginning and end of summer; (2) the bottom panels show that the crust index values in high-geodiversity hillslopes are higher in the periods at the beginning and end of summer. Note also that the values of both indices are lower on 30 September 2019, because this time of the year is after the long dry summer and before the rainy season.

Figure 14 shows the SMC spatial distribution of the study area calculated by the OPTRAM method for 3 June (Panel a) and 23 September (Panel b) during 2019. Overall, the SMC is higher at the high-geodiversity hillslopes in accordance with the CWCI results.

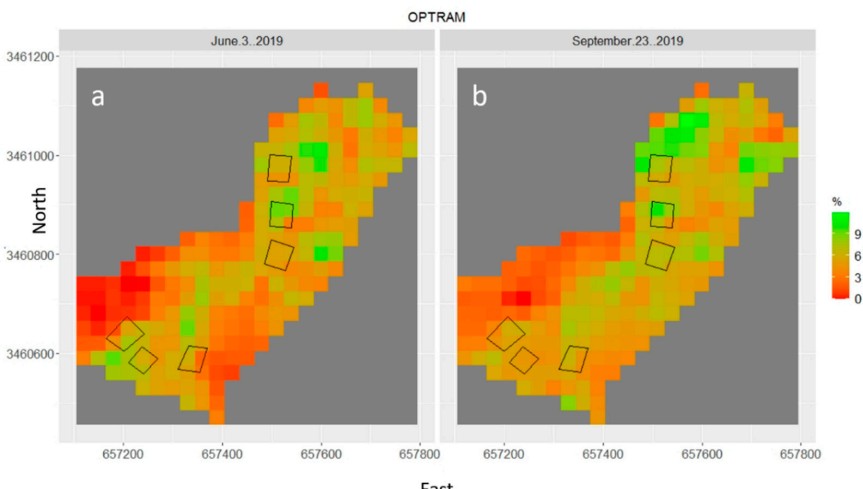

**Figure 14.** SMC spatial distribution calculated by OPTRAM for the same dates as in Figure 13. Map (**a**) presents the SMC in the June 2019, while Map (**b**) presents the SMC in October 2019. The SMC is obviously higher in the high-geodiversity hillslopes, in agreement with the CWCI results.

## 4. Discussion

Semiarid shrublands experience consistent water scarcity, which affects vegetation productivity [3,8]. Under these stressed conditions, soil patches with relatively high SMC can become a haven for vegetation species, and therefore they have drawn the attention of researchers in the current literature [58,61]. Field observations revealed that areas with high geodiversity are characterized by substantially greater SMC than areas of low geodiversity [15]. Nevertheless, the observations in previous studies were sporadic, both in space and time, and were focused on data from the wet season only. Utilizing data from LANDSAT 8 and images from the recently launched VENµS satellite provided us with more comprehensive evidence about the effect of geodiversity and rock fractions on SMC seasonal dynamics in this semiarid environment. Validation of the LANDSAT 8-based OPTRAM data with on-site measurements proved that high-geodiversity areas are moister (Figures 5 and 6), and the results of the CWCI and crust index, based on the high spatial and temporal resolution VENµS satellite images, showed similar results. The high correlation between the soil water (OPTRAM) and water in vegetation (CWCI, shown Figure 11) is typical in water-limited systems where there is a positive feedback between biomass and soil water. More water means extended roots and also increase in infiltration under the vegetation patch. These two feedbacks result in "in phase" soil–water distributions, which is common in mathematical models for vegetation patterns that include the infiltration feedback [62].

The new findings from the VENµS data show that the SMC is greater and almost stationary in the dry season in high-geodiversity hillslopes; additionally, in these plots, biocrusts were found to be more developed. The developed biocrust in the high-geodiversity hillslopes, as predicted from the VENµS data, can be explained by the higher SMC, coupled with the soil textural effect of a higher silt and clay content [46] that enhances biocrust growth [62]. These results are also supported by the findings of Assouline et al. [9] and Nejidat et al. [63], who showed that biocrust is more developed in high-geodiversity areas, where the high stoniness cover increases SMC via runoff.

The main findings show that SMC is greater in areas of high geodiversity. In particular, large differences in SMC between the two geodiversity levels were observed during the wet season, while in the dry season, only small differences were observed. According to these results, we propose the conceptual model schematically presented in Figure 15. High geodiversity positively affects the biocrust layer, which decreases infiltration and thus increases runoff in the high-geodiversity area. The increase in runoff is also due to an abundance of rock fragments on the surface. In addition, high geodiversity negatively affects the water loss through the lower evaporation rate from bare soil (the interspace between the shrubs patches) due to stones and more developed biocrust cover. Therefore, areas with high geodiversity contain high SMC. In the dry season, when rainfall is absent, evaporation and transpiration are the main water-related processes that occur. The fact that the SMC in the high-geodiversity plots remains high implies a low evaporation rate. This hypothesis is supported by previous works in semiarid environments [12,13]. During the wet season, the contribution of rainfall to SMC is considerably greater in the high-geodiversity hillslopes. This is closely related to the decrease in infiltration into the deep soil layers in the bare soil patches due to the stone cover and the more developed biocrust cover, which contribute to runoff generation from the bare soil to the vegetation patches [44,64,65]. This effect is known as the infiltration feedback, which is one of the main feedbacks for vegetation pattern formation in semiarid ecosystems [7,66]. The infiltration feedback is attributed to physical or biogenic crusts that develop on bare soil and cause an infiltration contrast between the vegetation patch and the surrounding area, and thus enhance overland water flow from the bare soil to the vegetation patch. Since the difference in SMC between the two hillslope types is more pronounced in the wet season, the infiltration feedback is more dominant than the decrease in evaporation in the dry season due to stoniness and biocrust cover. Another possible factor that can decrease the evaporation from the high-geodiversity hillslopes is the lower land surface temperature (LST) in summertime (Figure S1). The LST in high-geodiversity hillslopes is lower by ~1–2 °C than the low-geodiversity hillslopes. The mitigation of soil-water loss in the high-geodiversity hillslopes

allows vegetation to survive prolonged droughts [14,17]. The proposed conceptual model needs to be validated with further specific measurements of evaporation and runoff generation over the year in both hillslope types.

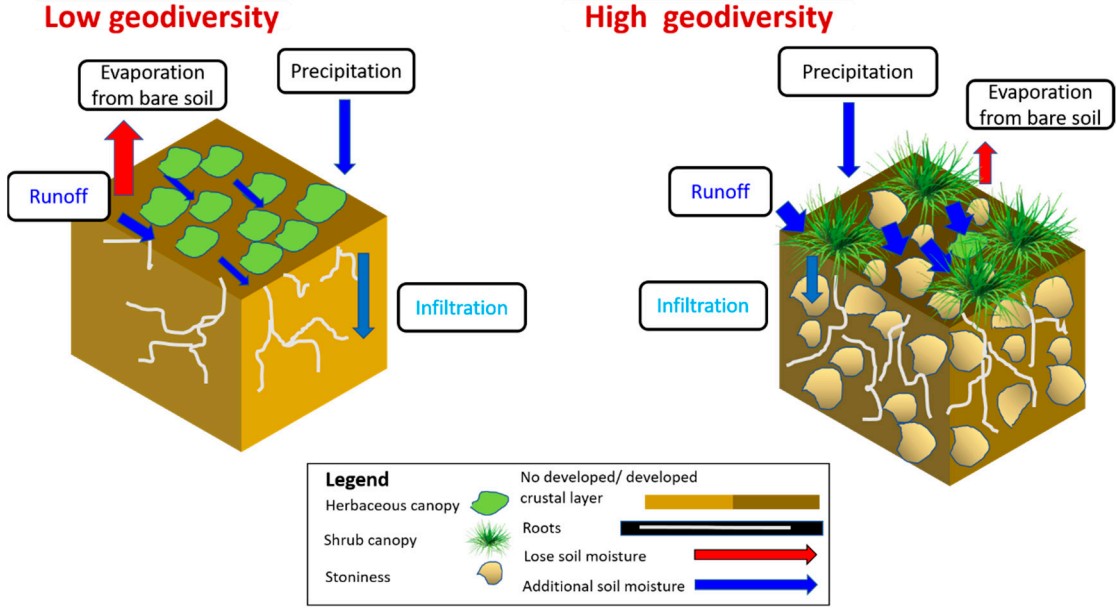

**Figure 15.** Conceptual scheme illustrating the geodiversity effects on soil moisture storage: evaporation, runoff, and infiltration in the two hillslope types. High geodiversity positively affects the biocrust layer, which decreases infiltration and thus increases runoff in the high-geodiversity area. The increase in runoff is also due to surface rock fragments at the surface. In addition, high geodiversity negatively affects the water loss through the lower evaporation rate from bare soil (the interspace between the shrubs patches) due to stones and biocrust. Therefore, areas with high geodiversity contain high SMC. The length of the arrows indicates the relative intensity of the process.

Another insight of this study is the validation of OPTRAM as a reliable proxy for estimating SMC. In previous works, OPTRAM-based SMC data were only tested in humid environments, with high vegetation cover densities, in which NDVI values were significantly higher than 0.3 [36]. In contrast, semiarid shrubland ecosystems are characterized by less rainfall and by bare soil patches [15,46]. Unexpectedly, despite the high cover percentage of bare soil, the OPTRAM-based SMC data showed a high correlation with the field measurements ($R^2$ = 0.9565, *p*-value < 0.01). This result implies that even with low NDVI values, OPTRAM can still be used to estimate SMC. This method can assist in monitoring SMC conditions in semiarid environments that are infrequently and incompletely assessed by field measurements. Therefore, OPTRAM is an efficient optional method to assess the SMC in water-limited environments, and it can help to monitor vegetation mortality and desertification processes.

## 5. Conclusions

The results presented in this work may lead to the following conclusions:

(1) Areas of high geodiversity retain greater SMC than areas of low geodiversity. These results are in agreement with previous evidence of an ameliorative effect of geodiversity at coarser spatiotemporal scales, using field measurements in limited locations.

(2) Predictions of SMC dynamics using OPTRAM-based time series from LANDSAT 8 data showed that SMC is substantially greater in the high-geodiversity hillslopes during the wet season.

(3) The high correlation between OPTRAM's SMC predictions and field measurements shows that the use of this remote sensing methodology to monitor SMC using LANDSAT 8 in semiarid environments is reliable.

(4) Our results reveal a high correlation between OPTRAM estimates of SMC and the CWCI computed from VENμS images, representing vegetation canopy water content. Therefore, the high spatiotemporal resolution of VENμS images can also be used to monitor moisture at the patch level in drylands together with the calibration with OPTRAM.

(5) OPTRAM calculated from LANDSAT 8 images is a better way to estimate SMC by using remote sensing. However, in case that a finer spatial resolution is needed, CWCI from VENμS can be used to estimate SMC

(6) The biocrust index applied to VENμS images has shown that the areas of high geodiversity have a more developed biocrust coverage during the summer, which may decrease the evaporation rate from the bare soil.

(7) A better understanding of the effects of geodiversity due to the presence of stones and rock fractions on the durability of dryland ecosystems to prolonged droughts will enable the designing of new management practices, to better address the predicted climatic change scenarios.

**Supplementary Materials:** The following are available online at http://www.mdpi.com/2072-4292/12/20/3377/s1, Figure S1: Land surface temperature maps of the studied plots for 2019-06-03 and 2019-09-23; Table S1: The list of LANDSAT 8 images for OPTRAM calculations; Table S2: The list of VENμS images for CWCI and crust index calculations.

**Author Contributions:** V.D., T.S. and H.Y. were responsible for writing the paper and analysis of the remote sensing data; H.Y. and I.S. were responsible for obtaining the field measurements and controlling the experiments in the LTER Sayeret Shaked Park; I.S. was responsible for the acquisition of funding. All authors have read and agreed to the published version of the manuscript.

**Funding:** This research was supported by the Israel Science Foundation (ISF), grant number 1260/15.

**Acknowledgments:** We thank Dolev Termin and Doron Haramati for their fruitful comments on the manuscript. We also thank the Israel Ministry of Science and Technology for their permission to use VENμS images and the USGS for the use of the LANDSAT 8 images.

**Conflicts of Interest:** The authors declare no conflict of interest.

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
