# Peer review of "Using LANDSAT 8 and VENµS Data to Study the Effect of Geodiversity on Soil Moisture Dynamics in a Semiarid Shrubland"

_remotesensing, doi:10.3390/rs12203377_

Round 1

Reviewer 1 Report

Dear authors,

Thank you very much for very interesting reading. Before the publishing of this manuscript, I have some comments, suggestions and few corrections that should be made.

First of all, I was very interested what do you mean by geodiversity. In the Introduction you refer some attributes, but is this a standard definition or it is just subjective selection of few abiotic factors in one cited work? Can you please add why you have chosen those factors that refers and represents geodiversity?

In the Instruction for authors, the citation in the main text should be corrected (e.g. [1-3], [4-7], etc.).

The graphical characterization of the study areas is very good. But it looks like the localities with high and low geodiversity are pretty close. Can you state how far away are these “two groups” from each other? I am not very convinced that these two groups are in soil characteristics (soil depth) and vegetation diversity that different.

Why did you measure SMC at an exact 7.6 cm depth? Is it connected with LANDSAT 8 depth measurement?

The third graph in Figure 6 miss the description, I suppose 2017-2018 labelling should be added.

Please, also check in the instruction for authors how and where to put the description for tables and figures. Correct and delete one description for Figure 11. Before publishing of this manuscript, please carefully check the whole text as there are few mistakes.

I believe that many soil properties (biological, chemical and other physical) influence SMC. Unfortunately, you do not provide this information and therefore, we cannot be sure why high geodiversity is characterized by greater SMC. As you mentioned in the discussion part, it is connected with the soil textural effect (I would say also with this), but you have not measured those parameters. Can you add at least that information and develop the discussion part?

Overall, I liked this manuscript very much, it was interesting to read. After the corrections, I would recommend to publish your work within this journal.

Reviewer 2 Report

August 02, 2020

Manuscript: ‘Using LANDSAT 8 and VENµS data to study the effect of geodiversity on soil moisture dynamics in a semiarid shrubland’

The aim of this paper is to assess the effect of geodiversity, expressed by degrees of stoniness, on regulating the seasonal dynamics of soil moisture content (SMC) by using LANDSAT 8 and VENμS images in a semiarid region in the northwestern Negev Desert, Israel. The optical trapezoid model (OPTRAM) based on normalized difference vegetation index (NDVI) and land surface temperature (LST) data was used to estimate SMC, whereas in situ measurements from field campaigns were used as benchmark data. This is a relevant topic lies within the scope of the MDPI remote sensing journal. The article is well organized and neatly written with the appropriate scientific content. Based on the above, I support the publication of this manuscript, but only after a minor revision.

********************************

Title: it fits perfectly the paper content. 

Abstract: it is quite adjusted to the paper content, but its aim should be clearly presented in line 17.  

Introduction: it provides sufficient background and includes relevant references on the estimation of SMC from remote sensing data and its interaction with ecosystem’s physical and biotic components in semiarid environments. Objectives and the novelty are clearly presented.

Materials and Methods: the study site, datasets and methods are adequately described, but I think that this section could be significantly improved adding a flow chart with the different methods described in text, highlighting inputs, applied techniques, and outputs so that readers could understand this section easier.  

Line 212: fix ‘…the highest 95% of the transformed reflectance (r) data for the maximal border.’

Line 273: fix ‘…Ordinary Least Squares (OLS).’

Results: these are clearly presented.

Lines 282-287: I think these lines summarize the aim of this work; so, re-write and move from here to Abstract.

Lines 296-298, the caption of Figure 3: for consistency, move this text to the previous paragraph ‘During the five-year period, the SMC for the high geodiversity hillslopes is higher than that in the low geodiversity hillslopes’

Lines 308-310, the caption of Figure 4: ‘The line that connects the two boxplots represents the difference in the mean SMC values that is statistically significant (p-value < 0.05).’ Where is that line? Please, verify.

Line 315, Table 1: to avoid duplicity, delete Table 1, because this finding was mentioned in previous paragraph (lines 300-301).   

Lines 333-335, the caption of Figure 5: move this text to the previous paragraph ‘The difference in SMC values between the hillslopes is larger in the winter and much smaller during the dry summer months.’

Lines 347-350, the caption of Figure 6: move this text to the previous paragraph ‘The maximum peak of soil moisture is in February, in agreement with the SMC measurements based on LANDSAT 8 data. Note that the SMC is always larger in the high geodiversity hillslopes.’

Line 355, the caption of Figure 7: move this text to the previous paragraph ‘The correlation between the two measurement types is very good.’

Line 371, the caption of Figure 8: move this text to the previous paragraph ‘It is clearly larger in the high geodiversity hillslopes.’

Line 376, Table 2; line 380, Table 3: to avoid duplicity, delete Tables 2 and 3, because this finding was mentioned in previous paragraph (lines 359, and 361-362).   

Line 385, the caption of Figure 9: move this text to the previous paragraph ‘The correlation between the two methods is quite good, which means that the CWCI calculated from VENμS data can be used to estimate SMC.’

Tables 4 and 5: to avoid duplicity, delete Tables 4 and 5, because this finding was mentioned in previous paragraph (lines 390 and 393).   

Figure 11: replace ‘x’ and ‘y’ with ‘East’ and ‘North’, respectively

Discussion: the content of this section is clear and concise and is in line with results.

Line 463: in this context, I would suggest: ‘This hypothesis is supported by previous works…’

Lines 490-495, the caption of Figure 12: move this text to the previous paragraph ‘High geodiversity positively affects the biocrust layer, which decreases infiltration and thus increases runoff in the high geodiversity area. The increase in runoff is also due to surface rock fragments at the surface. In addition, high geodiversity negatively affects the water loss through the lower evaporation rate from bare soil (the interspace between the shrubs patches) due to stones and biocrust. Therefore, areas with high geodiversity contain high SMC.’

Conclusions: these are clear and concise and are in line with results.

Linea 712, Appendix B: Fix it.

Reviewer 3 Report

First of all, I am sorry to say that I have many concerns with this manuscript. I believe that the authors did many things to explore the impact of geodiversity on soil moisture dynamics; however, looking at the results from a hydrological aspect, I have trouble understanding and agreeing with the assertions that the authors have made.

To begin, even though it is crucial to consider the slope and facing the slope when it comes to hillslope hydrology, the authors have not mentioned about anything about these subjects. To account for the impact of geodiversity, they should have a fixed slope and facing of the slope. For example, the amount of radiation, amount of available water for ET, LAI, types of species, and location of the slope are major characteristics that determine the variability of water flux. With no water-limited or LAI-limited assumptions, the upper slope would appear to have higher ET than SM due to the higher Rnet. Meanwhile, the absolute soil moisture (SM) value would appear to be lower over the upper slope due to the occurrence of more ET on the previous day and no rainfall on the present day. Furthermore, water from the upper slopes subsidizes the lower slope and increases SM values. Thus, investigating a few study areas without any constraints related to other critical hillslope factors and asserting that geodiversity is positively related to SM does not make sense to me. In my opinion, this kind of study should be done on a laboratory scale, not on a field scale. Please refer to the following references:

Band, L. E., Patterson, P., Nemani, R., & Running, S. W. (1993). Forest ecosystem processes at the watershed scale: incorporating hillslope hydrology. Agricultural and Forest Meteorology, 63(1-2), 93-126.

Lu, N., & Godt, J. W. (2013). Hillslope hydrology and stability. Cambridge University Press.

Huff, D. D., O'Neill, R. V., Emanuel, W. R., Elwood, J. W., & Newbold, J. D. (1982). Flow variability and hillslope hydrology. Earth Surface Processes and Landforms, 7(1), 91-94.

Second, the authors used the same ground measurement of SMC data as [17]; however, I cannot agree that the SMC data [17] obtained from the field is reliable because very few data were collected for this study area.  Even though it is essential to calibrate SMC data measured from the TDR sensor using gravimetric SMC data, there was no information regarding the calibration of TDR SMC data. In addition, the authors asserted that the sensing depth is 7.6-cm; however, this is not true. The electrode length is 7.6-cm. Please refer to the following references:

Cosh, M. H., Jackson, T. J., Bindlish, R., Famiglietti, J. S., & Ryu, D. (2005). Calibration of an impedance probe for estimation of surface soil water content over large regions. Journal of Hydrology, 311(1-4), 49-58.

Cosh, M. H., Ochsner, T. E., McKee, L., Dong, J., Basara, J. B., Evett, S. R., ... & Sayde, C. (2016). The soil moisture active passive Marena, Oklahoma, in situ sensor testbed (smap‐moisst): Testbed design and evaluation of in situ sensors. Vadose Zone Journal, 15(4), 1-11.

Third, I cannot agree that low ET and high runoff resulted in high SM. Without knowing the exact amount of ET and runoff fluxes, I have doubts about the entire manuscript. The authors should at least investigate the Budyko Framework. I personally feel that the authors have not fully discussed their results based on the basic physics of hydrology. Please refer to the following references:

Gurtz, J., Baltensweiler, A., & Lang, H. (1999). Spatially distributed hydrotope‐based modelling of evapotranspiration and runoff in mountainous basins. Hydrological processes, 13(17), 2751-2768.

Hirschi, M., Mueller, B., Dorigo, W., & Seneviratne, S. I. (2014). Using remotely sensed soil moisture for land–atmosphere coupling diagnostics: The role of surface vs. root‐zone soil moisture variability. Remote Sensing of Environment, 154, 246–252.

Famiglietti, J. S., & Wood, E. F. (1991). Evapotranspiration and runoff from large land areas: Land surface hydrology for atmospheric general circulation models. Surveys in Geophysics, 12(1-3), 179-204.

Reviewer 4 Report

The manuscript entitled "Using LANDSAT8 and VENμS data to study the effect of geodyversity on soil moisture dynamics in a semi arid shrubland" by V. Dublin, T. Svoray, I. Stavi, H. Yizhaq presents the application of the OPTRAM methodology for the derivation of soil moisture from short wave spectral bands from satellite sensors (here Landsat8 and VENμS) at high resolution (30 m & 5 m) over a semi arid region of Israel and its analysis in relation to degree of soil stoniness. The study reveals a higher soil moisture content especially during wet season in areas of high geodiversity from both remote sensing methods and ground sensors. Another indicator (crust index) that can be derived from hyperspectral remote sensing data is then associated to the analysis to explain the possible reason of this result.

The topic is of interest to the readers of Remote Sensing, the study is well referenced and the objective is very clear, the different steps are relatively well described. However, I find that at some places important information is missing, and graphical results are redundant.

Comments:

1) 2.1 study sites: An exact composition of the plots should be given, and a picture of each could be given. This is to better understand the differences in composition, relief orientation, ...

2) Figure 2: Latitude and longitude should be added.

3) l.195-196: The change from Band 7 to Band 6 is proposed because of a higher correlation. Could you give an explanation of this result?

4) l.282-289: This paragraph should be moved to a  sub section of methodology.

5) The results  of Landsat8 presented in Fig 3, 4, Table 1, Fig 5, Fig 6 & Fig 7 are highly redundant, one could expect to concatenate the findings in one or two graphical results. For example, the time series of both OPTRAM-LandSAT (high & Low geodiversity) compared to in-situ, with R² score (and RMSE) and potentially the boxplot of Fig 4 as a separate illustration (but without the table 1, the numbers can be added to the figure). Please revise the caption of figures (Fig 4 caption is misleadin: it is not a comparison of hillslopes and I do not see the line connecting the two box plots; Fig 3: what are the continuous lines compared to the points?).

6) VENμS data are used for SMC estimation in a completely different manner. Why is it so? The use of correlation with OPTRAM to imply a good estimation of SMC from another index (CWCI) is not very correct (l.365  & Fig 9). A better method would involve the direct validation with in-situ data, which I find should be shown in the manuscript.

7) Again, the caption of Fig 9 should be more explicit on what we see in the graph. What are the red points and the shaded area?

8) Caption of Fig 11 is repeated, please remove one.

9) Please insert an image of OPTRAM Landsat8 for SMC for comparison, as in Fig 11.

10)  What do you suggest to monitor SMC? Do OPTRAM from Landsat8 and CWCI give the same accuracy for SMC over the area? An explanation of the comparison and recommendation for use could be given in the text, as a conclusion.

11) As I pointed out in a previous comment, Conclusion (4) may be far stretched if you do not validate directly CWCI against in-situ. If it would be done, it could give more credit to transfering the total explanation with the Crust Index towards the soil moisture analysis.

Reviewer 5 Report

Using LANDSAT 8 and VENµS data to study the effect of geodiversity on soil moisture dynamics in a semiarid shrubland

Vladislav Dubinin, Tal Svoray, Ilan Stavi, Hezi Yizhaq

Submitted to Remote Sensing ref 897485

General comments

This paper includes a number of interesting items. However some of them concern remote sensing methodologies while others are related to observed soil moisture issues, in such a way that one is not sure what is the main issue considered in the paper.

using both remote sensing information and ground truth, the authors present results which indicate a significant difference in soil moisture behaviour and vegetation water content, depending on the geodiversity, which I take to be the presence of stones in the soil layer. My problem here is that I do not understand the physical reason why the effect occurs with the indicated trend (that is: more stones result in increased soil moisture). Probably an explanation is that I am not familiar with semiarid situations; still, clarification is needed and cannot be completely left to references.

The critical explanation may be hidden in lines 465-69 which mention the "infiltration feedback".  This remains however rather enigmatic; moreover, the relevance of one of the references is problematic. The conceptual scheme proposed on figure 12 does not help much.

While the authors stress the role of biocrust, it is not clear whether such a layer contributes to increased or decreased soil moisture.

On the whole, in y opinion the paper should be better focused, and the physical insight in observed behaviours made simpler;

Detailed comments

L037 1 introduction

01 L077 I cannot find the meaning of "run on" in this context

02 L111 what is the order of magnitude of the "hill slope scale"?

03 L120-124 does biocrust increase dryness? Yes, according to L121, as "biocrust allows high evaporation from the soil due to low albedo"

L136 2. Materials and Methods

04 L147-153 it is surprizing that areas with so vastly different thicknesses of soil layer are found close one to each other. Can an explanation be offered?

05 L144-45 Is the slope a critical component in the present study, or might similar results be obtained for flat terrain? I am puzzled by the insistence on slopes.

In addition, I was unable to find a definition for "loessial calcic xersol"

06 L147-153 Also, how do you define quantatively the stoniness and how do you estimate it?

07 L185 220: this follows closely ref(30)

08 L238 what is a "well correction"?

09 L257-58 "biological crust cover, which can decrease evaporation" seems to contradict text on line 121 above

10 L264: According to ref 34 this index seems to work provided cyanobacteria are present. Is it the case here?

11 deleted

12 L276-277 I fail to understand "the temporal trend is explained by a horizontal curve to the X-axis"

L280 3. Results

13 L282-284: state the purpose of the paper. But has not this been done earlier (lines 124-128)?  Yet the formulations are somewhat different.

14 L290-92: indeed, according to figure 3, the SMC for high geodiversity cases is higher. On the other hand, the evidence for seasonal dependence is not convincing.

15 L294: indications along the X axis do not help to separate clearly the rainy versus non rainy seasons.

15 L308: I do not see the line you are talking about

16 L340-42: while one hopes a fair correlation between remote sensing and field measurements, the correlation reported here indeed very high. Is it typical of field/LANDSAT correlation for similar (semiarid) conditions?

17 L360-61 I have some doubt about the minimum in August. This is just the empirical result of applying a Loess method. From the spread of the dots, it seems bold to go beyond a linear fit; unless of course you are able to propose a physical interpretation for a minimum…

18 L400 401: This sentence is unclear for me.

19 L407 low rather than law!

20 L455-56 I do not understand this at all. But this may be due to the fact that I am not familiar with semi-arid sceneries. To me, the effect of runoff it that the water disappears far away, instead of seeping in the ground toward the root zone. Visibly, the reasoning is quite different here, perhaps even completely opposite. But why?

21 L460-472; Well I believe I understand the role of biocrust, inasmuch as it reduces evaporation rate. On the other hand, I still do not understand how a soil with plenty of rocks and gravel may be able to store more water.

21bis L469: please explain the relevance of reference (60)

22 L487 Does not the rather good correlation between the OPTRAM and CWCI indexes deserve a discussion? While it suggests that in semiarid conditions the water contents in soil and vegetation are closely related, this may not be quite true, and anyway needs further analysis and interpretation.

23 L489; figure 12: I do not understand in the inserted legend the item "no developed/developed crustal layer": no corresponding colour code seems present in the figure.

23bis L489: Moreover, I do not understand why the runoff arrows point inside. I would naturally have drawn them the other way around.

L498 5. Conclusions

24 L500-03; This is indeed established, provided that the method for sorting high versus low diversity areas is made explicit and shown to be robust. However, the main difficulty encountered by the present reviewer is that he finds this result (more stones increase the SMC) counterintuitive and hardly understandable.

25 L510-13 As mentioned above, a discussion of the physical reasons for this good correlation would be highly welcome.

26 L514-16 This conclusion does not seem consistent with what the paper said above concerning the crust (I mean figure 10 and comments)

Round 2

Reviewer 4 Report

The revised version of the manuscript "Using LANDSAT 8 and VENµS data to study the effect of geodiversity on soil moisture dynamics in a semiarid shrubland " takes most of the reviewers comments in consideration. I am statisfied by the authors replies to the comments and the way they are implemented in the revised manuscript. From my point of view, I still find redundancy (and maybe sub-optimal way) in presenting the results of figures 5 to 9, but all the material is now well described and the manuscript easy (and nice) to read.